# A Signaling Game of Family Doctors and Residents from the Perspective of Personalized Contracted Service

**DOI:** 10.3390/ijerph191710744

**Published:** 2022-08-29

**Authors:** Zhiqiang Ma, Jialu Su, Hejun Pan, Mingxing Li

**Affiliations:** 1School of Management, Jiangsu University, Zhenjiang 212013, China; 2Affiliated Hospital of Jiangsu University, Zhenjiang 212001, China

**Keywords:** competency, family doctor, personalized signing service, signaling game

## Abstract

The role of the family doctor contracted service system in China’s medical and health system is increasing day by day. However, with the steady increase in contracting coverage, the phenomenon of “signing up but not contracting” has become common; to improve the current situation, the personalized signing service model has been strongly advocated. To promote the smooth implementation of the personalized contracted service model with family doctor competency as its core, this study used the signal game model to analyze the market equilibrium state of the signing service model. The results of this analysis reveal the following: (1) The camouflage of the number of contracts leads to distortion of the signal effect and to market failure, that is, the cost of competency camouflage is the primary factor affecting the equilibrium of contracted services. (2) The incompleteness of contracted services leads to quantity but not quality in the contracting market, that is, the payment of personalized service packages, the value-added utility of personalized services, and service gaps are the key factors that affect the decision-making behavior of the public. With this knowledge in mind, a compensation incentive mechanism that matches the competence level of the family doctor should be established, the formulation of contracted service agreements should be improved, and the participation of family doctors and residents should be encouraged, while the promotion of personalized contracted services should be enhanced and relevant supporting measures should be improved.

## 1. Introduction

In recent years, with the in-depth reform of China’s medical system, the family doctor contracted service system with general practitioners as the core has been of great significance to improving urban and rural residents’ access to medical care [1]. Especially at present, while the COVID-19 pandemic is spreading, the role played by primary healthcare institutions is increasingly prominent, and the family doctor contracted service is an important guarantee for the role of primary healthcare institutions. Since the adoption of the Guidance on Promoting Family Doctor Contracted Service in 2016, family doctor contracted service has been fully implemented in China. According to the Blue Book on Medical Reform: Report on the Development of China’s Medical Reform (2020), by the end of 2018, 320 million people were covered by contracted services for key populations in China, with a coverage rate of 71.3% [2]. It is evident that the construction of the family doctor contracted service system in China has been widely recognized by society and has achieved positive results.

However, although the contracted coverage rate has steadily increased, compared with more developed countries, the contracted service of family doctors in China is still in its infancy and is more complete in the form of administrative instructions [3]. Due to the complex and changing policy environment of healthcare reforms, the generally low motivation of family doctors, the information asymmetry between doctors and patients, and the incompleteness of contracting service agreements, among factors, it is easy to create a series of problems such as blindly pursuing the number of contracts, signing without an agreement, moral hazard, and adverse selection in the policy implementation process [4]. Research demonstrates that the provision of personalized contracted services oriented to residents’ needs plays a positive role in improving the willingness and satisfaction of subscription services [5]. However, as the subject of personalized contracted service, family doctors’ generally low competency levels greatly hinder the implementation and promotion of personalized contracted service. The question of how to further promote personalized contracted service by improving the competency of family doctors, and thus promote residents’ willingness to use contracted services, has become one of the urgent issues to be solved.

At present, most studies have focused on the construction of the family doctor competency model [6] or empirically examined the constraints of family doctor contracted services from a micro perspective [7]. However, few studies have analyzed the behavioral motivation of family doctors with different competency levels in personalized contracted service and its impact on residents’ contract strategy selection. Based on this lack of research, this paper adopted a signaling game approach to construct a game model between family doctors and residents concerning the contracted service with family doctors’ competency levels as the key signal. By analyzing the equilibrium solution of the model, this study analyzed the possible dilemmas in the implementation of the personalized contracted service model by combining specific parameters and, finally, proposes corresponding countermeasures from the perspective of the government.

## 2. Context

### 2.1. Family Doctor Competency

Since McClelland put forward the concept of competency in 1973 [8], the study of competency has become a popular topic, and scholars have different views on the definition of competency, with various discussions continuing into the present. Regarding the competency of family doctors, Patterson et al. argued that as the family doctors’ care for patients is transferred from hospital to community, the corresponding competency should shift from a “helping model” to a “business model” to adapt to the whole health system [9]. Hong et al. proposed a competency model including primary healthcare, community disease management, and prevention according to the working characteristics of family doctors [10]. Lu et al. found that basic medical service ability and interpersonal skills are the key factors in evaluating a family doctor’s post competency [11]. Based on the background of “Internet+” family doctors’ service, Ni et al. developed a scale of assessment for family doctors’ serviceability in China [12]. In conclusion, constructing a perfect competency model for family doctors to strengthen the construction of a family doctor team and improve the medical service system is of great significance [13].

In recent years, in response to the issue of how to enhance the competency of family doctors, more scholars have focused on the remuneration mechanism based on the competency of family doctors. In the family doctor contracted service, the State Administration of Traditional Chinese Medicine highlighted in 2018 that no less than 70% of the contracting service fee should be used as part of the family doctor team’s remuneration, and each family doctor should be assessed according to their number of contracts and service quality, reflecting the distribution principle of high performance and high remuneration. Therefore, as one of the main sources of income for family doctors, the contracted service fee has a significant incentive effect on family doctors’ competency. Wang et al. noted the significant incentive effect of paid services on family doctors’ contribution behavior [14]. Wordsworth et al. [15] and Whalley et al. [16] revealed that the contracting system placed more emphasis on performance-based pay and quality incentives, promoting job flexibility and significantly improving job satisfaction and participation of family doctors. In the current situation in China, the remuneration of family doctors is mainly related to their working age, educational background, and post [17]. This type of remuneration design has certain drawbacks. On one hand, family doctors have low social recognition [18] and their remuneration is far lower than those of specialists. If we do not consider the differences in competency between individuals, especially for young people, salary is not directly proportional to cost of time and energy, which will lead to low attractiveness of the family doctor’s position, making it difficult to retain talents, and to the family doctor lacking initiative and enthusiasm in learning on the job. On the other hand, this will also lead to unreasonable salary costs for government departments. Therefore, the salary incentive is the key to enhancing the competency of family doctors [19].

In addition, the relationship between family doctor competency and residents’ willingness to enroll has also attracted extensive academic attention. A study by Liang et al. showed that the most important factor that residents are concerned about regarding the family doctor service is the level of medical skill [20]. The investigation and analysis by Wang et al. showed that the abilities and service attitudes of family doctors are important factors affecting residents’ willingness to sign contracts [21]. Wang and Sun noted that the diagnosis and treatment environment, as well as the diagnosis and treatment ability of primary hospitals are the key factors affecting patients’ satisfaction and loyalty [22]. Therefore, family doctor competency is closely related to residents’ willingness to register.

In summary, for family doctors, the contracted service fee, as an important component of their salary income, has a clear economic incentive to encourage family doctors to improve the quality, competency, and value of their service. For residents, family doctor competency is an important factor influencing their willingness to sign contracts.

### 2.2. The Family Doctor Contracted Service

Since the implementation of the family doctor contracted service policy, various contracted models have been formed in practice and exploration across the country, but the phenomenon of signing but not contracting has still not been sufficiently improved. Many scholars have explored the reasons for the stagnation of the contracted service policy. For example, some scholars discovered that the conflict of interests among health administrations, primary healthcare institutions, family doctors (teams), and patients is an important reason for the policy stagnation [23]. Some scholars also noted that the basic service package designed at this stage is limited by the price and cannot meet the personalized needs of different groups, and that designing individualized contracted service packages with the support of government policies can not only increase the enthusiasm of family doctors but also improve the residents’ willingness to use contracted services [24]. Chen Bin et al. found that with the increase in the coverage rate of personalized contracted services, residents’ satisfaction and sense of access also increased [25]. Meanwhile, compared with the basic service package, the contracted service fee of the personalized service package has a more obvious compensation effect on the remuneration of family doctors and team members [26]. Based on this discovery, the successful implementation of the personalized contracted service model can not only motivate family doctors to improve their competency but also increase the satisfaction of contracted residents.

A review of the existing literature reveals that the remuneration incentives are the key to enhancing the competency of family doctors, while personalized services are the fundamental demand of residents. Linking contracted service to the capitation-based medical insurance compensation mechanism can motivate family doctors from the aspect of economic benefits and encourage them to attract more residents to sign contracts by improving their own competencies. On one hand, family doctors can receive a more lucrative salary by signing up with residents for higher-paying personalized service packages; on the other hand, with strong financial support from the government, residents can sign up for personalized service packages to meet their own health management needs at a lower cost. This is an effective way to improve both GP competency and resident satisfaction. However, there are certain obstacles in the implementation process of this plan. For example, family doctors, as signal senders, release signals of competency that do not match their abilities to receive more lucrative salaries, as a way to attract more residents to sign up for personalized service packages. As signal receivers, residents cannot accurately know the type of competency (strong or weak) of family doctors, which leads to the fact that family doctors with weak competency match more residents who sign up for personalized service packages, and thus the gap between reality and the ideal, due to differences in residents’ perceptions, leads to lower satisfaction with contracted services. Therefore, this paper constructs a game model based on competency signals to explore the difficulties and countermeasures of the implementation of the personalized contracted service model, aiming at stimulating the improvement of family doctors’ competencies by expanding the personalized contracted service package, and then stimulating residents’ willingness to use the contracted service through the improvement of family doctors’ competencies.

## 3. Model Construction and Equilibrium Analysis

### 3.1. Theoretical Mechanism

In the process of implementing the family doctor contracted service system, the stakeholders involved are the health administration department, primary healthcare institutions, family doctors (teams), and residents. As the direct suppliers and demanders of contracted services, family doctors and residents, respectively, are the main research objects of this paper. According to the analysis of the literature, there is a correlation between family doctor competency and residents’ willingness to sign up. Therefore, based on incentive theory and service management theory, this paper constructs the influence path of “family doctor competency→family doctor contracted service,” from the perspective of salary incentives and personalized service, to explore the key factors affecting the promotion of personalized contracted service.

### 3.2. Model Construction

A signaling game is a dynamic game with incomplete information. A typical signaling game consists of two players: the sender and the receiver. As the family doctor and the resident are two parties with asymmetric information, there is a sequence of actions between them in the process of implementing contracted services, so a signaling game is suitable for solving the research problem in this paper. Assuming that the game revolves around contracted services between family doctors and residents of the area, it has been shown that, in the healthcare service market, patients are unable to assess the service quality of doctors due to a lack of professional knowledge, and there is a large information asymmetry between the two [27], which in this paper signifies that residents only know the a priori probability of family doctor competency and the number of contracted personalized service packages, and it is difficult to confirm the actual competency of family doctors. Therefore, referring to existing studies that took the number of comments in the online consultation service as a basis for judging the doctors’ service levels [28], this paper took the number of signed personalized service packages as a key signal for residents to judge the competency levels of family doctors. When the number of contracts exceeds the average level, family doctors will be judged as having a high competency level; on the contrary, if the number of contracts falls below the average level, family doctors’ competency level will be judged as low.

The specific game process is as follows: Firstly, nature (N) determines the family doctor type with a prior probability t∈T, the type space $T=\left\{t_{H},t_{L}\right\}$, and $t_{H}$ and $t_{L}$ represent high-competency and low-competency family doctors, respectively, with a prior probability $P\left(t_{H}\right)=\alpha$, $P\left(t_{L}\right)=1-\alpha \;\left(\alpha \in \left(0,1\right)\right)$. Secondly, family doctors observe their own type t and then select the signal from the signal set to transmit; in the signal set $n\in \left\{n_{H},n_{L}\right\}$, $n_{H}\;$and $n_{L}$ represent higher and lower contracted quantities, respectively. Finally, according to the competency signal n transmitted by the family doctor, the residents use the Bayesian rule to modify the prior probability P(t) to obtain to which type of posterior probability P(t|n) the family doctor belongs, and then select their own strategies $S=\left\{s_{1},s_{2}\right\}$, where $s_{1}$ and $s_{2}$ represent the signing of a personalized service package and basic service package, respectively.

Regarding the parameter design for the signal recipient—the resident: The total cost for the resident to sign up for the personalized service package is $C=C_{1}+C_{2}$, where $C_{1}$ is the cost of investigation, that is, the cost paid by the resident to find the actual competency level of the family doctor, such as making inquiries with people who are familiar with the situation, and $C_{2}$ is the actual cost paid by the resident for signing up for the personalized service package. The total cost for residents to register for the basic service package is zero, and access is free upon registration. This paper assumes that residents are rational economic individuals, and considering that signing up for the basic service package is free, they all choose to register under the strong advocacy of the local government. Regarding the utility, for residents who sign up for the personalized service package, if they correspond to a high-competency family doctor, the utility they obtain includes the basic utility $V_{b}$ brought by the basic public health service and the value-added utility $V_{a}$ brought by satisfaction with personalized services; on the contrary, if they correspond to a low-competency family doctor, the value-added utility obtained by the residents is $\lambda V_{a}$ (0<λ<1), under the condition that the family doctor does not disguise their own competency. λ represents the service gap between a high-competency family doctor and a low-competency family doctor, where the gap decreases with λ. Under the condition that the family doctor disguises his or her competency, a successful disguise results in the resident obtaining value-added utility $V_{a}$, but residents will not be able to obtain value-added utility if the disguise fails, which also means that the gap between reality and the ideal based on individual cognitive bias will bring some losses to residents.

Regarding the parameter design of the signal sender—the family doctor: In terms of cost, the marginal cost of basic public healthcare service is $C_{b}$, and the marginal cost of personalized healthcare service for family doctors with a high or low competency level is, respectively, $C_{L}$ and $C_{H}$, where $0<C_{L}<C_{H}$; the higher the competency, the lower the marginal cost of service. In addition, a family doctor with a low competency level may act speculatively to disguise his or her competency in order to obtain more benefits, thereby sending a false signal of high competency to the resident. The camouflage cost of transmitting this false signal is $C_{0}$. This speculative behavior generally means that family doctors do not focus on improving their own competency but use “being signed up,” “signing up for,” and “false signing up” based on China’s unique social structure of acquaintances such as “favor and face” to increase the number of contracts for their own personalized service packages [21]. With regard to benefits, family doctors will receive funding $M_{1}$ for basic public healthcare services regardless of the service package for which they register, which is mainly funded by the basic medical insurance fund and government subsidies. In addition, family doctors who sign up for the personalized service package will also receive payment $\eta C_{2}$ for the personalized service package, of which 0<η<1, which means that the self-paid portion of the residents who register for the personalized service package will be included in the family doctor’s salary in proportion. This will motivate family doctors’ efforts to a certain extent, but at the same time, it also breeds opportunistic behavior, a key point of this study. Finally, if the resident who registers for the personalized service package is satisfied with the family doctor, the family doctor will receive an additional benefit W, which mainly represents the improvement of the family doctor’s reputation. More residents may sign up for the personalized service package with them in the future. For family doctors with high competency, the probability that residents are dissatisfied with them is very small and may be ignored. For a family doctor with low competency, the probability of success in disguise is γ, 0<γ<1. If the disguise succeeds, the family doctor will receive the same additional benefits as a family doctor with high competency; if the disguise fails, the residents will not be satisfied with them. At this time, they will not only be unable to obtain additional benefits but will also be subject to certain penalties π.

The benefits to the family doctor and the resident under different conditions are shown in Figure 1. The benefits to the family doctor’s income are specifically expressed as $UA_{1}=M_{1}+\eta C_{2}+W-C_{b}-C_{L}$, $UA_{0}=M_{1}-C_{b}$, $UA_{2}^{\prime }=\gamma \left(M_{1}+\eta C_{2}+W-C_{b}-C_{H}-C_{0}\right)+\left(1-\gamma \right)\left(M_{1}-C_{b}-C_{H}-C_{0}-\pi \right)$, $UA_{2}=M_{1}+\eta C_{2}+\lambda W-C_{b}-C_{H}$, and $UA_{0}^{\prime }=M_{1}-C_{b}-C_{0}$. The benefits to the resident are specifically expressed as $UB_{1}=V_{b}+V_{a}-C$, $UB_{0}=V_{b}$, $UB_{2}^{\prime }=\gamma \left(V_{b}+V_{a}-C\right)+\left(1-\gamma \right)\left(V_{b}-C\right)$, and $UB_{2}=V_{b}+\lambda V_{a}-C$.

### 3.3. Equilibrium Analysis

Assuming that the probability of a false signal of competency from a family doctor is θ (θ∈[0,1]), there are $P(n_{H}|t_{H})=1$*,*
$P(n_{L}|t_{H})=0$*,*
$P(n_{H}|t_{L})=\theta$*,* and $P(n_{L}|t_{L})=1-\theta$. According to the Bayesian rule, the posterior probability of the residents is $P(t_{H}|n_{H})=\frac{\alpha }{\alpha +\left(1-\alpha \right)\theta }$, $P(t_{L}|n_{H})=\frac{\theta \left(1-\alpha \right)}{\alpha +\left(1-\alpha \right)\theta }$, $P(t_{H}|n_{L})=0$, and $P(t_{L}|n_{L})=1$.

#### 3.3.1. Separating Equilibrium

Since competent family doctors will not release a signal for a low number of contracts, there is only one condition left for a possible separating equilibrium strategy for family doctors, in which they send a signal for a high number of contracts, and family doctors with low competency send a signal for a low number of contracts. Under this condition, there are $P(n_{H}|t_{L})=0$ and $P(n_{L}|t_{L})=1$.

(a) When residents receive a signal from family doctors that the number of contracts is high ($n_{H}$), the expected benefit of their choice to sign up for the personalized service package is $EB_{1}=P(t_{H}|n_{H})UB_{1}+P(t_{L}|n_{H})UB_{2}^{\prime }=UB_{1}=V_{b}+V_{a}-C$; the expected benefit from choosing to sign up for the basic service package is $EB_{0}=P(t_{H}|n_{H})UB_{0}+P(t_{L}|n_{H})UB_{0}=V_{b}$. When $EB_{1}>EB_{0}$, the residents’ optimal strategy is to sign up for the personalized service package. We can solve $V_{a}>C$. In this case, competent family doctors always choose to send signals with a high number of contracts and will not deviate from the equilibrium path. As far as the reality is concerned, the proportion of residents who sign up for personalized service packages at their own expense is low, and the value of the services covered far exceeds the signing cost of residents. However, the premise for residents to obtain full access $V_{a}$ is to fully understand and actively use the contents of the personalized service package; at the same time, personalized service packages should be designed for different groups of people, to improve residents’ willingness to use them. Conversely, if residents do not understand the personalized service package or the personalized service package cannot meet their needs, then $EB_{1}<EB_{0}$, where the corresponding signing cost is higher than the value-added utility obtained. In this scenario, the residents’ optimal strategy is to sign up for the free basic service package.

(b) When residents receive a signal from family doctors that the number of contracts is low ($n_{L}$), the expected benefit of signing up for a personalized service package is $EB_{2}=P(t_{L}|n_{L})UB_{2}=V_{b}+\lambda V_{a}-C$; the expected benefit of signing up for a basic service package is $EB_{0}^{\prime }=V_{b}$.

When $EB_{2}>EB_{0}^{\prime }$, the residents’ optimal strategy is to sign up for personalized service packages. We can solve $V_{a}>\frac{C}{\lambda }$. Residents will choose to sign up for personalized service packages only when the service gap between various types of family doctors is small. When residents choose to sign up for personalized service packages, the family doctor’s benefit on the equilibrium path is $UA_{2}$, and the benefits on the disequilibrium path are $UA_{2}^{\prime }$ and $UA_{0}^{\prime }$. If the benefits are satisfied by $UA_{2}>UA_{2}^{\prime }$ and $UA_{2}>UA_{0}^{\prime }$, the family doctor does not deviate from the equilibrium path. We can solve $C_{0}>max\left\{C_{H}-\eta C_{2}-\lambda W,\;\left(\gamma -\lambda \right)W-\left(1-\gamma \right)\left(\eta C_{2}+\pi \right)\right\}$. At this point, the family doctor and the resident reach a separating equilibrium, which is classified in this paper as Equilibrium Ⅰ, and the equilibrium solution is $\left\{\left(n_{H}\;n_{L}\right)\;\left(s_{1}\;s_{1}\right)\;P(t_{H}|n_{H})=1\;P(t_{L}|n_{L})=1\right\}$. In this equilibrium, all types of family doctors choose to release true signals and residents sign up for personalized service packages.

When $EB_{2}<EB_{0}^{\prime }$, the optimal strategy for residents is to sign up for a basic service package, at which point $V_{a}<\frac{C}{\lambda }$, indicating that the residents are reluctant to sign up for a personalized service package when the service gap is large, that is, when the value-added utility received by the residents cannot cover the cost paid. At this point, the benefit of family doctors on the equilibrium path is $UA_{0}$, and the benefits on the disequilibrium path are $UA_{2}^{\prime }$ and $UA_{0}^{\prime }$. When $UA_{0}>UA_{2}^{\prime }$ and $UA_{0}>UA_{0}^{\prime }$, a separating equilibrium is achieved, and the solution is $C_{0}>\gamma \left(\eta C_{2}+W+\pi \right)-C_{H}-\pi$. Since $max\left\{\gamma \left(\eta C_{2}+W+\pi \right)-C_{H}-\pi \right\}=\eta C_{2}+W-C_{H}$, when $C_{0}>\eta C_{2}+W-C_{H}$, the family doctor and the resident reach a state of separating equilibrium, which is classified as Equilibrium Ⅱ, and the equilibrium solution is $\left\{\left(n_{H}\;n_{L}\right)\;\left(s_{1}\;s_{2}\right)\;P(t_{H}|n_{H})=1\;P(t_{L}|n_{L})=1\right\}$. In this equilibrium, family doctors with either high or low competency levels all release true signals, corresponding to contracted personalized service packages and basic service packages, respectively.

#### 3.3.2. Pooling Equilibrium

The possible pooling equilibrium is one in which all types of family doctors release the signal of a high number of contracts, where there are $P(n_{H}|t_{L})=\theta =1$ and $P(n_{L}|t_{L})=0$. When the residents receive the signal $n_{H}$, they choose to sign up for the personalized service package with the expected benefit of $EB_{3}=P(t_{H}|n_{H})UB_{1}+P(t_{L}|n_{H})UB_{2}^{\prime }=\left\{\gamma +\left(1-\gamma \right)\alpha \right\}V_{a}+V_{b}-C$, or the basic service package with the expected benefit of $B_{0}=P(t_{H}|n_{H})UB_{0}+P(t_{L}|n_{H})UB_{0}=V_{b}$.

When $EB_{3}>EB_{0}$, the solution is $\left\{\gamma +\left(1-\gamma \right)\alpha \right\}V_{a}>C$, which is also $\alpha >1+\frac{C-V_{a}}{\left(1-\gamma \right)V_{a}}$. In this case, if $V_{a}<C$, the above formula is impossible. In other words, when the contracted cost is greater than the value-added utility, the residents do not choose to sign up for the personalized service package; in the case where the cost of residents signing up for a personalized service package is less than the value-added utility, when the probability of the family doctor’s success in disguise decreases, the proportion of doctors with actual high competency in the total family doctor team subsequently increases, and the optimal strategy of residents for this scenario is to sign up for the personalized service package. When residents choose to sign up for personalized service packages, high-competency family doctors will not deviate from the equilibrium path; they will always be willing to send signals $n_{H}$ of a high number of contracts. The benefit of low-competency family doctors on the equilibrium path is $UA_{2}^{\prime }$, and the benefits on the disequilibrium path are $UA_{2}$ and $UA_{0}$. Only when $UA_{2}^{\prime }>UA_{2}$ and $UA_{2}^{\prime }>UA_{0}$ will the family doctor not deviate from the equilibrium path, and the solution is $0<C_{0}<min\left\{\left(\gamma -\lambda \right)W-\left(1-\gamma \right)\left(\eta C_{2}+\pi \right),\;\gamma \left(\eta C_{2}+W+\pi \right)-C_{H}-\pi \right\}$. In this scenario, the family doctor and the resident reach a pooling equilibrium, which is classified as Equilibrium Ⅲ, and the solution is $\left\{\left(n_{H}\;n_{H}\right)\;\left(s_{1}\;s_{1}\right)\;P(t_{H}|n_{H})=1\;P(t_{L}|n_{H})=1\right\}$ or $\left\{\left(n_{H}\;n_{H}\right)\;\left(s_{1}\;s_{2}\right)\;P(t_{H}|n_{H})=1\;P(t_{L}|n_{H})=1\right\}$. The implication of this equilibrium is that all family doctors release the signal $n_{H}$ and residents sign up for the personalized service package. In this case, the low-competency family doctors trick residents into signing up for a personalized service package by inventing a false number of contracts to receive a handsome salary. This is a “free-riding” behavior, which means that rather than gaining the trust of residents by improving competency, family doctors instead take shortcuts to obtain a pay increase.

When $EB_{3}<EB_{0}$, $\left\{\gamma +\left(1-\gamma \right)\alpha \right\}V_{a}<C$, there is $\alpha <1+\frac{C-V_{a}}{\left(1-\gamma \right)V_{a}}$. In this case, if $V_{a}<C$, then the above formula must be established. In other words, when the cost of signing up for the personalized service package is greater than the value-added utility, the optimal strategy for the residents is to sign up for the basic service package; if $V_{a}>C$, the proportion of high-competency doctors in a family doctor team decreases if a family doctor has a greater probability of success in disguise. More family doctors mask themselves instead of improving themselves, and the optimal strategy for residents is to sign up for the basic service package. When residents choose to sign up for basic service packages, high-competency family doctors always choose the equilibrium path of a high number of contracts. For low-competency family doctors, the benefit of the equilibrium path of a high number of contracts is $UA_{0}^{\prime }$, and the benefits of the disequilibrium path of a low number of contracts are $UA_{2}$ and $UA_{0}$. When a low-competency family doctor chooses to release the signal $n_{L}$, if the resident chooses to sign up for the basic service package, the family doctor’s income is $UA_{0}$. Since $UA_{0}^{\prime }<UA_{0}$ is obviously established, the family doctor will definitely deviate from the equilibrium path, and there will be no pooling equilibrium in this scenario. Therefore, there are two necessary conditions for achieving a pooling equilibrium: first, when the family doctor deviates from the equilibrium path to choose to release the signal $n_{L}$, the resident chooses to sign up for the personalized service package; secondly, when the resident chooses to sign up for the personalized service package, the benefit of the family doctor on the disequilibrium path is less than that on the equilibrium path. In this case, the family doctor has no motive to deviate from the equilibrium path, and the two sides reach a pooling equilibrium. Satisfying $\{\begin{array}{c} UB_{0}<UB_{2}\\ UA_{2}<UA_{0}^{\prime } \end{array}$, the solution yields $\frac{C}{V_{a}}<\lambda <\frac{C_{H}-C_{0}-\eta C_{2}}{W}$, at which point the two sides achieve a pooling equilibrium, classified as Equilibrium Ⅳ, and the equilibrium solution is $\left\{\left(n_{H}\;n_{H}\right)\;\left(s_{2}\;s_{1}\right)\;P(t_{H}|n_{H})=1\;P(t_{L}|n_{H})=1\right\}$. In this equilibrium, all types of family doctors choose to release the signal $n_{H}$, while residents choose to sign up for the basic service package; when residents receive the signal $n_{L}$, they instead choose to sign up for the personalized service package, which is a phenomenon of adverse selection.

In summary, in the contracted service focused on family doctor competency, four equilibria will develop between family doctors and residents, as shown in Table 1. Combined with the above analysis, it can be found that the cost of disguise $C_{0}$, the payment for the personalized service package acquired by family doctors, the value-added utility of the personalized service $V_{a}$ acquired by residents, the service gap between high-competency family doctors and low-competency family doctors, and the proportion of high-competency family doctors in the family doctor team are the most important factors that affect the decision-making behavior of the subject. In order to facilitate the discussion, the parameters were abstracted into two states of large (high) and small (low). Then, a specific analysis of the real-life dilemmas that hinder the implementation of the personalized contracted service model was conducted in relation to the parameters in the model.

## 4. Discussion

This paper analyzed the behavioral motivations of family doctors with different levels of competency in personalized contracted services and their influence on residents’ choice of contracting strategies. The main findings are as follows.

It is worth noting that family doctor competency directly affects residents’ willingness to sign up, and in this paper, competency was expressed through the number of contracts. Family doctors can disguise their competency through false numbers of contracts. Whether family doctors camouflage competency mainly depends on the cost of disguise. When the cost of disguise $C_{0}$ is low, the whole family doctor contracted service market is in a state of pooling equilibrium, and all types of family doctors release the signal $n_{H}$ and consider themselves to have a high competency level. This is deserved for high-competency family doctors, while low-competency family doctors deceive residents. Specifically, aiming at Equilibrium Ⅲ, low-competency family doctors gain residents’ trust by disguising the number of personalized service packages for which they register, encouraging residents to sign up for personalized service packages with them. This “free-riding” behavior allows low-competency family doctors to crowd the market of personalized contracted services, which in turn reduces residents’ satisfaction and utilization of personalized contracted services and hinders the overall promotion of the personalized contracted service model. At the same time, a large number of low-competency family doctors will follow suit and also take shortcuts instead of improving their own competencies to obtain rich remuneration, and the number of family doctors sending false signals in the contracted service market will gradually increase. When the proportion of low-competency family doctors exceeds a critical value ($1-\alpha >\frac{V_{a}-C}{\left(1-\gamma \right)V_{a}}$), the equilibrium strategy evolves from Equilibrium Ⅲ to Equilibrium Ⅳ, and the phenomenon of “adverse selection” occurs. This means that low-competency family doctors are more popular with residents than high-competency family doctors. In this scenario, high-competency family doctors are unable to attract residents to sign up and improve the quality of personalized contracted services through their own competency advantages. In this situation, high-competency family doctors are not sufficiently motivated to provide services, narrowing the service gap with low-competency family doctors, which in turn leads to a lower cost of disguise for low-competency family doctors and further worsens the dysfunctional state of the contracted service market. At this time, residents’ satisfaction and trust will be greatly reduced, which is not conducive to the promotion of the contracted service policy of family doctors. In reality, the lower cost of disguise has led to a watered-down sign-up rate in more primary care institutions. For example, a 2018 survey conducted by the Guangdong Provincial Health and Planning Commission revealed that many community health service centers had increased their sign-up rates by “pulling heads”, but there was a large “shortage” of family doctors and follow-up services could not be kept up. Without enhancing the competency of family doctors, it is an avoidance to focus on the design of the contracted service package and the contracted coverage rate.

Additionally, the family doctor’s competency must be matched with the service quality. When the cost of disguise $C_{0}$ is high, family doctors choose to release real signals, so residents have a clear idea of the actual competency level of each family doctor. In this case, the family doctors’ service quality becomes an important factor influencing residents’ willingness to sign up for personalized service packages. In this paper, the family doctor’s service quality was reflected in the value-added utility and service gap. Analysis showed that when the value-added utility of the personalized service package signed up for by the residents and the service gap are small, regardless of the competency of the family doctor, residents will choose to sign up for a personalized service package only if the cost $C_{2}$ is very low. In other words, residents are reluctant to pay more for a personalized contracted service package when the basic package and the personalized contracted service package bring almost the same value to them. However, when the contracting cost is low and the value-added utility of personalized contracted service is small, residents are not willing to use the personalized contracted services, and it is very easy to sign up but not contract, which not only consumes the government subsidy but also does not motivate family doctors to improve their competency through remuneration. Based on the hypothesis of “rational-economic man” and the mechanism of profit-seeking behavior, family doctors choose to release a truthful signal when the cost of disguise is high and the actual remuneration is low. The family doctor, as the party with the information advantage, can take speculative behavior to reduce the marginal cost of the service, while the residents, as the party with the information disadvantage, do not understand the potential value of the contracted service and intuitively believe that a low contracted cost means a low service value, so they give up receiving services. According to Equilibrium Ⅱ, the ideal equilibrium can only be achieved if both family doctors and residents are motivated, and family doctors take the initiative to improve their competency and widen the service gap, and if at the same time, residents actively use the contracted services and thus increase the perceived value-added utility.

Based on the above analysis, it was found that family doctors’ competency and residents’ willingness to use contracted services are closely related and mutually reinforcing, with personalized contracted services playing an important mediating role. On one hand, the higher payment for the personalized service package provides an incentive for family doctors to improve their competency in order to receive more remuneration. On the other hand, the higher competency level of family doctors increases the perceived value added by residents who sign up for personalized service packages, which in turn leads to increased satisfaction and greatly enhances the willingness of residents to sign up for the personalized contracted service.

### 4.1. Practical Enlightenment

Through in-depth discussion of this study, the following important enlightenments are obtained.

#### 4.1.1. Improving Service Fee Allocation Mechanism for Personalized Contracted Services

In order to improve family doctors’ initiative and enthusiasm in the contracted service, a remuneration incentive mechanism matching the level of competency of family doctors should be established. First, from the equilibrium analysis, in order to achieve the ideal state of separating equilibrium, the payment for personalized service packages obtained by family doctors should be high. Some studies have shown that the availability of residents’ out-of-pocket payments is the main factor affecting the income of family doctors [26]. Therefore, raising the amount of contracted service fees that can be allocated to individual family doctors or teams is one of the effective incentives. Second, in view of the information asymmetry between government departments and family doctors, it is difficult for government departments to understand the actual competence level of family doctors. As rational economic people, family doctors often choose their own actions based on the maximization of utility. Therefore, in order to avoid the emergence of pooling equilibrium, the behavior motivation of family doctors should also be considered when designing the salary incentive mechanism. The patient selection and payment mechanism can be combined to form incentives for family doctors while also implementing constraints. The reason for this is that in the pay-per-head incentive mechanism, the establishment of reputation mechanism is one of the effective ways to restrain doctors’ behavior, which can reduce the possibility of family doctors’ disguised competence level to a greater extent, and reputation mainly comes from the feedback of contracted residents. Finally, in the personalized service process, the content beyond the agreed service is charged according to the diagnosis and treatment items, which can encourage family doctors to improve the service quality and establish their own unique competitiveness to attract residents to sign up for personalized service packages.

#### 4.1.2. Multi-Party Participation in the Development of Contracted Service Agreements

In the process of formulating the contracted service agreement, the government should take the lead and encourage family doctors and residents to participate, to fully consider the interests of all parties. For example, in the process of formulating personalized signed service agreements, on one hand, more refined service content must be designed and reasonably priced. Today, with the implementation of policies and more achievements, residents’ sensitivity to prices has decreased. Therefore, in the process of designing personalized contracted service packages, the personalized needs of residents should be prioritized to provide more refined services. Further, in addition to continuing to pay attention to the elderly, pregnant and lying-in women, the disabled, low-income groups and poor households with filing cards, young people and high-income groups who have a great demand for health management, and residents who know about family doctor signing services should also become the key coverage targets of the personalized signing service system, and differentiated pricing can be adopted to meet the needs of different groups. On the other hand, the high-quality service of family doctors is also a key factor to enhance the value-added utility of contracted residents. First, the premise of providing high-quality services is to form a supporting legal system for family doctor signing services, to ensure the completeness of signing service agreements and avoid moral hazard. Secondly, the training mechanism and performance appraisal mechanism of family doctors should be strengthened and improved to enhance the competence of family doctors.

#### 4.1.3. Increasing the Promotion of Personalized Contracted Services

In the publicity of personalized contracted service, family doctors should first go deep into the community to carry out contracted service policy publicity activities, explain the content, service mode, service package types, and significance of personalized contracted service to residents, spread the concept of contracted service to promote health management, and cultivate residents’ awareness of prevention and healthcare. Second, it is necessary to reasonably guide the residents’ expectations of the contracted services and avoid exaggerating the service contents, thus resulting in dissatisfaction caused by the gap between the residents’ psychological expectations and the reality. Finally, we should give full play to the influence of residents who have enjoyed the value-added effect of personalized contracted service, build a good reputation through word of mouth, and create a good publicity atmosphere in the community to trust and support the personalized contracted service of family doctors, to make relevant policies easier for residents to interpret.

## 5. Conclusions

This paper analyzed the equilibrium state of family doctors with strong and weak competency levels by constructing a signal game model and discussed the implementation dilemma of the personalized contracted service model with the relevant parameters in the game model. The findings suggest that the cost of disguise $C_{0}$, the payment for the personalized service package acquired by family doctors, the value-added utility of the personalized service $V_{a}$ acquired by residents, the service gap between high-competency family doctors and low-competency family doctors, and the proportion of high-competency family doctors in the family doctor team are key factors influencing residents to sign up for personalized service packages. This paper also put forward corresponding countermeasures and suggestions. The contributions of this study are as follows: In theory, most of the previous relevant literature used the empirical research method, which is limited by data availability and the scope of the survey; the research conclusions have certain limitations. This paper used signal game theory to explore the interaction process between family doctors and residents in the signing service, enriching the research content in this field and the application scope of signal theory. At the practical level, the previous research seldom focused on the analysis of personalized contracted service. The research conclusion of this paper has direct and important practical significance to promote the implementation of the personalized contracted service model with family doctor competency as the core, and to improve the national medical reform system.

## Figures and Tables

**Figure 1 ijerph-19-10744-f001:**
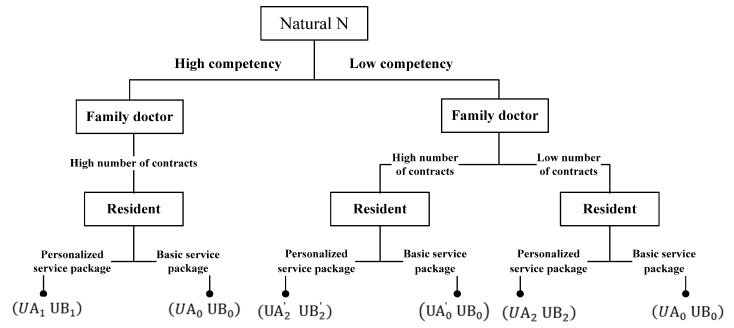
Signal game diagram of the family doctor contracted service.

**Table 1 ijerph-19-10744-t001:** Equilibrium states of the signaling game.

Cost of Disguise $\left(C_{0}\right)$	Payment for Personalized Service Packages by Family Doctors $\left(\eta C_{2}\right)$	Value-Added Utility of Personalized Services $\left(V_{a}\right)$	Service Gap (1−λ)	Proportion of High-Competency Family Doctors (α)	Equilibrium State
high	low	small	small		Equilibrium Ⅰ
	high	large	large		Equilibrium Ⅱ
low	low	small	small	large	Equilibrium Ⅲ
	low	small	small	small	Equilibrium Ⅳ

## Data Availability

Not applicable.

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
