# Peer review of "A Signaling Game of Family Doctors and Residents from the Perspective of Personalized Contracted Service"

_ijerph, 2022, doi:10.3390/ijerph191710744_

Round 1
Reviewer 1 Report
Interesting article that merits implementation taking into consideration competence and quality of service. However, the results may not be too clear to some readers. It may require some more detail explanation- too convoluted and would discourage readers as it concentrated too much on getting the formula correct. It would help if some examples are put in the formula to explain clearly.
Author Response
Point 1: Methods –Can be improved.
Response 1: Thank you for your suggestion. We have already introduced the signaling game in the Model construction section and pointed out the applicability of the method in this paper.
Point 2: Results – The results may not be too clear to some readers. It may require some more detail explanation- too convoluted and would discourage readers as it concentrated too much on getting the formula correct. It would help if some examples are put in the formula to explain clearly.
Response 2: Thank you for your valuable comment. We have made changes in the Discussion section to add more explanation to the results so that the reader can more easily understand the results of this paper. At the same time, we have given real-life examples to echo the results of this paper and increase the readability of the paper. Thanks again for your valuable comments, which are very important for the improvement of this paper.
Reviewer 2 Report
An interesting paper that discusses a novel approach to improving the family doctor model in China. The paper is very well structured and provides an in depth analysis of the game model related to the family doctor contracted service system. The outcomes of the game model and subsequent equilibrium analysis align well with the current issues identified in the literature and the forthcoming recommendations fill a gap highlighted by the authors in the introduction. Overall, a valuable theoretical paper that, if implemented, will improve health outputs and outcomes. Some minor points to consider below.
Line 29 - avoid using the term "raging". Something more academic would improve the flow of this sentence.
Line 48 - this sentence does not make sense. remove the first "implementation"
Line 65 - I would avoid using the heading literature review unless it was a systematic review with a full methodology attached. Perhaps use "Context" instead.
Perhaps including a formal discussion (add a heading) that includes the Practical Enlightenment section would provide more research focus. The conclusions are well described.
Author Response
Point 1: Line 29 - Avoid using the term "raging". Something more academic would improve the flow of this sentence.
Response 1: Thank you for your valuable comment. We have replaced the word and checked the whole paper for academic accuracy.
Point 2: Line 48 - this sentence does not make sense. remove the first "implementation".
Response 2: Thank you for your helpful suggestion. We have removed the first "implementation".
Point 3: Line 65 - I would avoid using the heading literature review unless it was a systematic review with a full methodology attached. Perhaps use "Context" instead.
Response 3: Thank you for your helpful suggestion. We have changed "Literature review" to " Context".
Point 4: Perhaps including a formal discussion (add a heading) that includes the Practical Enlightenment section would provide more research focus. The conclusions are well described.
Response 4: Thank you very much for this suggestion. We have carefully considered your suggestion and referred to some papers and have finally made minor changes to the structure of the paper. Firstly, we have divided the discussion and conclusion into two parallel sub-sections; Secondly, more content has been added to the discussion section to increase readability; Finally, the practical enlightenment has been condensed into three sub-points for the findings of this paper, with corresponding sub-headings so that the reader can clearly grasp the focus of the research.
Reviewer 3 Report
This manuscript shows a very important discussion from another perspective of medical care.
In the conclusion part, please, add summarized your conclusion in this paper.
Author Response
Point 1: In the conclusion part, please, add summarized your conclusion in this paper.
Response 1: Thank you for your valuable comment. We have summarized our conclusion in this paper.